# Spatiotemporal Distribution Patterns of Osthole and Expression Correlation of the *MOT1* Homologue in Cultivated *Angelica biserrata*

**DOI:** 10.3390/ijms262110746

**Published:** 2025-11-05

**Authors:** Kaidi Yu, Yuying Yang, Yuan Luo, Xiaogang Jiang, Jie Guo, Xiaoliang Guo

**Affiliations:** Key Laboratory of Biology and Cultivation of Herb Medicine, Ministry of Agriculture and Rural Affairs, Institute of Chinese Herbal Medicine, Hubei Academy of Agricultural Science, Enshi 445000, China; yukaidi@hbaas.ac.cn (K.Y.); yangyuying@hbaas.ac.cn (Y.Y.); 202430449@hbmzu.edu.cn (Y.L.); jiangxiaogang@hbaas.ac.cn (X.J.)

**Keywords:** *Angelica biserrate*, Osthole biosynthesis, *AbOMT1*, WGCNA, spatiotemporal metabolomics

## Abstract

Sustainable cultivation of *Angelica biserrata*, a medicinal species with a bioactive coumarin, Osthole, is hindered by inconsistent metabolite accumulation. To address this limitation, we integrated spatiotemporal metabolomics and transcriptomic analyses. Tissue-specific measurements revealed that root bark accumulates Osthole at 0.30 ± 0.15%, a concentration 11-fold higher compared to root pith and 15–30-fold higher compared to aerial organs. Over time, the Osthole content increased by 195% from September to December, with frost exposure further increasing the accumulation by an additional 170%. Germplasm screening identified an elite accession, AB-222, exhibiting 230% higher Osthole content compared to regional averages. Weighted gene co-expression network analysis identified a gene module strongly correlating with Osthole accumulation. Within this module, *AbOMT1* (*AB04G05077*), an *O-METHYLTRANSFERASE 1* (*OMT1*) homolog encoding an S-adenosyl methionine-dependent O-methyltransferase, was the top hub gene. *AbOMT1* expression reflected Osthole dynamics both spatially (three-fold higher in root bark vs. root pith) and temporally. Module functional analysis revealed significant enrichment in phenylpropanoid and monoterpenoid biosynthesis pathways. Our results suggest *AbOMT1* as a possible key molecular marker for Osthole accumulation, establish frost induction as a strong yield regulator, and suggest AB-222 as an elite germplasm resource.

## 1. Introduction

*Angelica biserrata* (R.H. Shan & Yuan) C.Q. Yuan & R.H. Shan represents a globally significant medicinal plant characterized by its phytochemical complexity, particularly in root tissues where diverse volatile and non-volatile metabolites have been identified. Comprehensive metabolomic analyses have classified its root extracts into 15 distinct chemical categories, with Osthole (7-methoxy-8-isopentenoxycoumarin) serving as a major bioactive constituent underpinning its pharmacological applications [1]. The biogeographical distribution of this species is governed by bioclimatic predictors such as annual precipitation gradients, which critically define regional cultivation suitability [2]. Economically, optimized extraction protocols, most notably solid-phase extraction coupled with HPLC-based quantification, enable cost-efficient Osthole isolation from *A. biserrata* with batch-to-batch consistency (RSD ≤ 4.6%) and industrial scalability [3,4]. These methodologies achieve exceptional purity yields exceeding 31.1% while demonstrating potent radical scavenging capacities, positioning Osthole as being suitable for applications beyond traditional medicine [4]. Nevertheless, safety considerations necessitate rigorous evaluation due to documented CYP3A4 enzyme inhibition by these extracts, which may influence co-administration with conventional therapeutics [5].

The pharmacological utility of Osthole spans anti-inflammatory, oncologic, neuroregulatory, and osteogenic domains. These effects are mediated through diverse molecular pathways. *TNF-α* suppression concurrent with *α7nAChR* upregulation mediates its anti-inflammatory efficacy, substantially ameliorating anxiety-like phenotypes and social withdrawal in experimental models via cholinergic pathway modulation [6,7]. In oncology contexts, Osthole exerts antiproliferative effects primarily through *PI3K*/*AKT*/*mTOR* pathway inhibition in retinoblastoma while demonstrating the capacity to reverse multidrug resistance mechanisms in solid tumors [8,9]. Neuroprotective actions through PPARα agonism ameliorate cognitive deficits, though clinical translation is constrained by limited blood–brain barrier permeability [6,10]. Additionally, osteogenic bioactivity mitigates osteoporosis phenotypes without inducing adverse effects typical of bisphosphonates [11], and promising agrochemical applications have emerged via contact toxicity and oviposition deterrence against major arthropod pests [12,13]. However, despite these therapeutic potentials, persistent spatial heterogeneity in Osthole accumulation (3.6–8.2× higher concentration in phloem versus xylem) compounded by seasonal fluctuations exceeding 300% significantly impairs production efficiency [14,15,16]. Current integrative strategies to address these limitations combine *MALDI*-*MSI*-guided harvesting protocols with genomic-based engineering of *POLYKETIDE SYNTHASE* (*PKS*) and *PRENYLTRANSFERASE* (*PT*) biosynthetic nodes to enhance content uniformity and sustainable resource utilization [17,18].

Despite substantial pharmacological characterization, agricultural implementation of *A. biserrata* is hampered by systematic research deficiencies regarding its industrial crop physiology. Critical knowledge gaps persist across several interdependent domains. Osthole’s heterogeneous distribution across root phloem versus xylem tissues and its minimal accumulation in leaves remain inadequately quantified, with metabolomic reports confirming 3.6–8.2× concentration gradients that significantly compromise harvesting efficiency [19,20]. Additionally, although developmental variations suggest seasonal fluctuations exceeding 300% in bioactive metabolites, field-validated temporal dynamics across growth cycles lack empirical verification, potentially leading to suboptimal harvest scheduling [7,21]. Furthermore, the molecular machinery governing biosynthesis remains largely unmapped, impeding targeted genetic enhancement [18,22]. Compounding these agricultural constraints, current multidisciplinary research primarily focuses on therapeutic mechanisms rather than production optimization [10,12]. This study addresses these industrial bottlenecks through integrated spatiotemporal metabolomics, transcriptomics, and environmental modeling. Our methodology aims to provide high-resolution mapping of Osthole accumulation patterns to enable precision harvesting while identifying transcriptional regulators of terpenoid/coumarin biosynthesis (e.g., *OMT* and *PT*) to advance molecular breeding for yield-stabilized cultivars adapted to specific bioclimatic conditions [2,23,24].

This study addresses critical industrial bottlenecks in *A. biserrata* cultivation through an integrated approach combining spatiotemporal metabolomics and transcriptomics. Our primary objective is to develop a comprehensive understanding of Osthole distribution and biosynthesis to optimize industrial cultivation practices. To achieve this, we map Osthole accumulation patterns across different plant tissues, characterize temporal metabolite dynamics throughout the growth cycle, identify environmental factors affecting biosynthesis, screen germplasm resources for elite chemotypes, and elucidate the molecular mechanisms governing Osthole production through weighted gene co-expression network analysis. Through these integrated approaches, we aim to establish optimal harvesting protocols and identify key molecular targets for genetic enhancement of this pharmacologically significant species.

## 2. Results

### 2.1. Hierarchical Accumulation of Osthole as a Functional Metabolite

To investigate the ecological functions of specialized metabolites, we performed tissue-specific quantification of six bioactive compounds in *A. biserrata* (Appendix A). This analysis revealed Osthole as the most differentially partitioned metabolite across root cortex, root core, petiole, and blade tissues, demonstrating maximal statistical variance (lowest *p*-value, *p* = 0.016) and the widest concentration range (MAX-MIN Ratio = 94.7) relative to other compounds via Tukey’s multiple comparison test (Appendix A). Such significant tissue heterogeneity suggests specific mechanisms governing Osthole biosynthesis or storage. Spatial mapping further identified a steep concentration gradient: root bark accumulated 0.3 ± 0.15% Osthole, which is 11-fold higher than root pith (0.028 ± 0.007%; *p* = 0.002, Student’s *t*-test) and 15–30-fold greater than aerial parts (petioles: 0.02 ± 0.005 mg/g, blades: 0.010 ± 0.002 mg/g) (Figure 1a,b). This bark-centric accumulation localized 83.6% of the total Osthole in underground organs, suggesting that its primary biological roles may involve belowground ecological interactions.

### 2.2. Metabolic Variability and Core Germplasm Identification

To delineate metabolic variability implicated in our metabolite profiling, we performed multivariate analysis on 12 *A. biserrata* individuals derived from a half-sibling family (Appendix A). Principal Component Analysis resolved 95.33% of compositional variance (PC1: 60.51%; PC2: 34.82%), isolating accessions AB-50, AB-222, and AB-SK2 as chemotypic outliers characterized by elevated PC1 scores (>1.5) and constrained PC2 values (0–0.5) (Figure 2a). These accessions consistently demonstrated the highest Osthole accumulation (Figure 2b). Coefficient of variation assessment further quantified heterogeneity: Osthole exhibited maximal variability (CV = 93.75%) compared to Osthenol (69.82%), Umbelliferone (64.7%), Columbianadin (57.8%), Isoimperatorin (33.56%), and Dihydrooroselol (27.11%) (Appendix A). This positions Osthole as the pivotal discriminator for germplasm screening, directly establishing AB-222 as the optimal core germplasm due to its exceptional chemometric stability.

### 2.3. Temporal Dynamics and Environmental Induction of Osthole Biosynthesis

Having established the distinct chemotypic variation among accessions (Figure 2), we further tracked temporal metabolite dynamics during rhizome maturation. Monthly monitoring of six bioactive markers in field-grown *A. biserrata* revealed a critical accumulation pattern for Osthole, which exhibited a content surge of 195% from September to November and continued rising, albeit at a reduced rate (17% increase), through December (Figure 3a; Appendix A). These three sampling points were strategically selected to capture key developmental stages in *A. biserrata*’s biennial growth cycle: the early dormancy phase (September), when vegetative growth ceases and resources begin redirecting to rhizomes; mid-dormancy (November), when secondary metabolite accumulation typically intensifies; and deep dormancy (December), which is the stage following frost exposure. Earlier time points were not examined as preliminary screenings of summer samples showed negligible coumarin levels that fell below reliable quantification thresholds, consistent with the resource allocation to aerial growth observed in *Apiaceae* perennials during vegetative phases.

Contrastingly, Osthole’s direct precursor Osthenol remained statistically unchanged (*p* > 0.05), while the upstream precursor Umbelliferone underwent a sharp decline (−50%) by November followed by a December surge to 40.16%, going above September baseline levels. To dissect environmental triggers, we analyzed nine accessions pre- and post-frost in December. An unexpected early frost event presented a valuable research opportunity to capture immediate metabolic responses within a 24-h window. Frost exposure induced a rapid 1.73-fold escalation in Osthole concentration across all lines (*p* < 0.05) (Figure 3b; Appendix A). The relative fold-change values displayed in Figure 3b (ranging from 1.63 to 1.95-fold across different accessions) highlight the remarkably consistent proportional increase in Osthole compared to the minimal relative changes in other metabolites, suggesting a specific frost-triggered biosynthetic shift favoring Osthole accumulation.

### 2.4. Transcriptomic Profiling and Co-Expression Network Analysis of A. biserrata

To identify molecular drivers of coumarin biosynthesis implicated by tissue-specific metabolite accumulation, we performed a transcriptome-wide investigation of the molecular mechanisms underlying coumarin biosynthesis. Total RNA sequencing of all corresponding metabolic samples generated comprehensive expression profiles (Appendix A). Principal component analysis confirmed robust sample stratification, revealing eight distinct clusters corresponding to tissue types (leaf, petioles, root bark, and root pith) and temporal variations across developmental stages (September, November, and December roots), with genetic divergence among 12 half-sibling individuals forming a separate cohort (Appendix A).

To elucidate regulatory networks governing coumarin accumulation, we executed WGCNA using our 33-sample FPKM dataset. Convergent scale-free topology fit indices and mean connectivity analyses identified a network construction threshold of power = 4, generating 44 co-expression modules (Appendix A). Integration of six key coumarin metabolite levels with expression clustering identified a gene module (colored purple in Appendix A) that strongly correlated with Osthole accumulation. The subsequent module–trait association analysis demonstrated that this module exhibited exceptionally strong positive correlations with Osthole (r = 0.81, *p* = 6 × 10^−4^) and Umbelliprenone (r = 0.74, *p* = 1 × 10^−6^; Figure 4a). Hierarchical clustering confirmed that co-expression patterns were tightly linked to Osthole accumulation (Figure 4b). Gene significance (GS) and module membership (MM) analyses within the purple module revealed significant relationships between Osthole levels and hub gene expression (Appendix A; Figure 4c), with topological visualization substantiating dense interconnection among key biosynthetic regulators (Appendix A).

### 2.5. Identification of AbOMT1 as a Central Hub Gene in the Osthole-Associated Co-Expression Module

To elucidate the molecular drivers underlying Osthole accumulation, this study integrated co-expression networks with metabolite correlation screening. The gene module strongly correlating with Osthole accumulation (r = 0.81, *p* = 6 × 10^−4^, colored purple in our visualization) was subjected to hub gene identification. Within this module, gene *AbOMT1* (*AB04G05077*), the homolog of *OMT1* (*METABOLIC O-METHYLTRANSFERASE 1*), emerged as a possible significant hub gene, demonstrating a robust correlation with Osthole content traits (gene significance, GS = 0.9162; module membership, MM = 0.8434). Functional annotation revealed that this gene encodes an S-adenosyl methionine (SAM)-dependent methyltransferase harboring the conserved IPR029063 SAM_binding domain, which is essential for O-methylation during coumarin backbone modification. Key co-expressed genes within the module included prenyltransferases (*AB10G00842*, *AB06G01999*, *AB07G04187*, and *AB11G02263*); *COUMARIN SYNTHASE* (*COSY*) regulating umbelliferone biosynthesis (*AB01G01375* and *AB06G01114*); and core phenylpropanoid biosynthetic genes (*PAL*, *C4H*, *4CL*, and *C2H*). All these genes are key components in the pathway of Osthole biosynthesis from phenylalanine (Figure 5a).

KEGG enrichment analyses revealed significant enrichment of genes from the purple module in the “phenylpropanoid biosynthesis” (ko00940, *p* = 1.87 × 10^−3^), “Diterpenoid biosynthesis” (ko00904, *p* = 2.81 × 10^−3^), and “terpenoid backbone metabolism” (ko00900, *p* = 1.67 × 10^−3^) pathways (Appendix A). GO enrichment analyses revealed significant enrichment of genes from the purple module in the “intramolecular oxidoreductase activity” (GO:0016860, *p* = 1.33 × 10^−3^), “lipid binding” (GO:0008289, *p* = 1.52 × 10^−2^) and “response to light stimulus” (GO:0009416, *p* = 4.21 × 10^−3^) pathways (Appendix A), indicating coordinated metabolic flux across these pathways. In contrast, KEGG and GO enrichment analyses of the gray module, which showed very low correlation with Osthole biosynthesis (r = −0.0027), revealed no enrichment in any pathways related to Osthole biosynthesis (Appendix A). This comparative analysis further supports the specificity of the purple module in Osthole biosynthesis regulation.

Validation via qRT-PCR using root samples representing different tissues, materials with varying Osthole contents, and different periods demonstrated that *AbOMT1* expression patterns strongly co-varied with Osthole accumulation. Specifically, expression in the root bark was three-fold higher than in the root pith, and expression progressively increased from September to December.

Crucially, the reliability of the underlying transcriptomic data was confirmed by a strong correlation (n = 434, |R| = 0.8052) between gene expression levels and qRT-PCR quantification for 31 selected genes across these diverse samples (Appendix A; Appendix A). Furthermore, a highly significant direct correlation (n = 26, |R| = 0.8786; Appendix A and Appendix A) was observed between the expression level of the key enzyme gene *AbOMT1* and Osthole accumulation intensity across these same sample sets (Figure 5b,c and Appendix A). These collective results position *AbOMT1* as a possible key regulatory factor in Osthole biosynthesis and as a prime candidate gene for metabolic engineering initiatives.

## 3. Discussion

### 3.1. Spatiotemporal Heterogeneity in Osthole Accumulation

The significant enrichment of Osthole in root bark (three-fold higher than in root pith) aligns with the established ecological defense roles of coumarins in subterranean tissues. This finding is supported by studies in the literature demonstrating that furocoumarins, including structurally related compounds, function as important natural biocides with specific “biodefense roles” in the roots of *Angelica dahurica*, where their accumulation is developmentally regulated [1]. Furthermore, complex coumarins in *Apiaceae* plants act as key regulators responding to environmental stimuli [25]. As derivatives of the phenylpropanoid pathway, this targeted enrichment represents a strategic defense mechanism against biotic stresses, providing a crucial physiological rationale for developing low-loss harvesting strategies focused on phloem-enriched tissues.

The pronounced increase in Osthole content (>150%) observed under seasonal changes and frost stress indicates a robust abiotic stress response mechanism. Cold stress activates the phenylpropanoid pathway, leading to enhanced accumulation of secondary metabolites like coumarins. This induction is mediated through signaling cascades involving endogenous hormones such as jasmonic acid and abscisic acid (ABA), which upregulate key biosynthetic and regulatory components [26]. Supporting this, cold stress has been shown to elevate phenylpropanoid derivatives, providing antioxidant protection against cold-induced oxidative damage [27]. Importantly, ABA signaling pathways are directly implicated in enhancing freezing tolerance [28], likely coordinating the observed metabolic shift towards coumarin accumulation. This suggests that controlled frost exposure could represent a viable agronomic strategy to boost Osthole yield in pharmaceutically relevant plants.

### 3.2. Expression of the AbOMT1 Correlates with Osthole Accumulation and Chemotype Stability

Elevated expression of *AbOMT1* correlated significantly with increased Osthole accumulation. This indicates a potential functional involvement of this SAM-dependent *AbOMT1* in Osthole biosynthesis. However, the specific catalytic role of *AbOMT1* in mediating the critical methylation step from osthenol to Osthole requires direct biochemical validation. The enzymes catalyzing this critical step of the coumarins biosynthesis pathway remain unidentified in *Angelica* species [29]. While *OMTs* like *AgOMT1* demonstrate functional methylation activity in related *Apiaceae* plants, such as *Anethum graveolens* for phenylpropene methylation [30], substrate specificity varies significantly across *OMTs*. Although we observed elevated expression of *AbOMT1*, its in vitro enzymatic characterization and confirmation of its ability to methylate specific intermediates like osthenol within *A. biserrata* are pending. Therefore, while our expression data strongly implicates *AbOMT1*, this evidence does not yet confirm it as the core catalytic enzyme for the final Osthole methylation, consistent with the knowledge gap highlighted in *A. decursiva* [29]. The observed expansion of *PRENYLTRANSFERASE* (*PT*) genes in *Angelica sinensis* for coumarin biosynthesis [15] further illustrates the complexity of the pathway and the need for functional validation of candidate genes like *AbOMT1* in specific species and steps.

This robust expression–metabolite correlation suggests constitutively elevated *AbOMT1* expression as an important factor underpinning the persistent, high-level Osthole production characteristic of the AB-222 chemotype. While we did not functionally assess specific regulatory polymorphisms as the causal mechanism for this differential expression, the correlative pattern itself aligns with established biological principles where stable metabolic phenotypes arise from genetically encoded differences in the regulation of key biosynthetic genes. For instance, stable variation driven by promoter polymorphisms controlling expression levels of key enzymes, such as *OsTPS1* in rice disease resistance [31], exemplifies this well-documented link between regulatory mechanisms and consistent metabolic traits. *OMT* expression modulation, as seen in sesquiterpene production in agarwood [32], further supports this paradigm within specialized metabolism. Future studies should prioritize fine-mapping and functional validation of regulatory elements within the *AbOMT1* locus in contrasting chemotypes, utilizing approaches successful in dissecting similar regulatory variation [31], to definitively elucidate the genetic basis for the persistent expression driving the economically valuable Osthole chemotype in AB-222.

### 3.3. Frost Stress Association with Osthole Levels and AbOMT1 Expression

Constitutively elevated *AbOMT1* expression, correlating with a high Osthole content, suggests inherent genetic regulation underpinning this metabolic phenotype. While our findings demonstrate frost-induced increases in Osthole levels, the question regarding the mechanism of frost induction of *AbOMT1* expression and subsequent ABA/ROS signaling as a contributor to Osthole accumulation remains open. No direct evidence has been established linking *AbOMT1* induction to cold stress as *OMT1* function is documented solely in the context of salt tolerance, where it enhances copper uptake to support ABA biosynthesis and compartmentalization of Na^+^ [33]. Consequently, a direct mechanistic link between frost stimuli and *AbOMT1* expression/activity in modulating Osthole remains speculative based on current evidence.

It is well-established that cold stress, including frost, engages ABA and ROS signaling as core conserved pathways across plants [34,35]. Activation of *OST1* kinase, induction of ABA biosynthetic genes like *NCED1* [35], accumulation of ROS [36], and crosstalk with other hormones such as JA [37] are fundamental mechanisms for cold acclimation, aiming to balance stress protection with resources for growth [38]. Given our data implicating *AbOMT1* in Osthole accumulation, it is plausible that perturbations in such core stress signaling networks could indirectly influence the expression or activity of biosynthetic genes or enzymes.

## 4. Materials and Methods

### 4.1. Plant Materials

Field specimens of *A. biserrata* were established on 20 March 2020 at an altitudinal range of 1200 m a.s.l. in Yesanguan Town, Badong County (30.632823° N, 110.336975° E), Hubei Province, a traditional cultivation region characterized by cambisol soil (pH 6.2; 0–20 cm depth: SOC: 22.7 g·kg^−1^, total N 159.2 mg·kg^−1^, Olsen-P 40.4 mg·kg^−1^, exchangeable K 193.5 mg·kg^−1^). Uniform one-year-old rootstocks showing vigorous growth (diameter ≥ 0.8 cm at root collar), absence of mechanical injury, and zero pathogenic symptoms were transplanted under non-fertilized cultivation at 75,000 plants·ha^−1^.

### 4.2. Sample Collection for RNA Sequencing and Metabolomic Profiling

An integrative multi-omics sampling approach was stratified across tissue types, genetic lineages, and seasonal phases in biennial *A. biserrata* plants. Three focal accessions (P1, P2, and P3) contributed leaf blade, petiole, root stele, and root periderm samples for parallel RNA/metabolome extraction. Ten half-sib families (AB-1, AB-12, AB-131, AB-133, AB-222, AB-24, AB-5, AB-50, AB-62, and AB-SK2) provided root tissues for comparative transcriptome–metabolome analysis. Temporal dynamics were captured through AB-182 root samples collected in September, November, and December 2022. Frost response metabolomics targeted root tissues of nine distinct accessions (AB-3, AB-5, AB-25, AB-55, AB-60, AB-67, AB-69, AB-70, and AB-92) sampled during pre-frost and post-frost periods, exclusively for metabolomic analysis.

### 4.3. Metabolite Extraction and Quantification

Samples dissected into roots, petioles, blades, root cores, and root cortices underwent thermal enzyme inactivation (105 °C, 15 min) followed by drying at 70 °C for 6 h until a constant mass was achieved. After pulverization through a 60-mesh sieve, targeted constituents were extracted using 50–85% ethanol via percolation or reflux. Simultaneous quantification of Dihydrooroselol, Osthole, iso-Osthenol, and Columbianadin was performed using HPLC on an Agilent Extend C_18_ column (250 mm × 4.6 mm, 5 μm). The mobile phase consisted of 0.1% aqueous formic acid (A) and acetonitrile (B) with a flow rate of 1.0 mL·min^−1^. The gradient elution program was as follows: 0–5 min, 25–30% B; 5–15 min, 30–40% B; 15–25 min, 40–55% B; 25–35 min, 55–70% B; 35–40 min, 70–80% B; 40–45 min, 80–25% B; and 45–50 min, 25% B for column equilibration. The column temperature was maintained at 30 °C with detection at 330 nm and an injection volume of 10 μL. Method validation confirmed linear responses within different concentration ranges, namely Dihydrooroselol (1.27–11.43 μg·mL^−1^, r ≥ 0.9996), Osthole (33.09–297.79 μg·mL^−1^), iso-osthenol (1.00–8.96 μg·mL^−1^), and Columbianadin (9.02–81.16 μg·mL^−1^), with average recoveries of 98.4–100.4% (RSD 0.90–1.72%, n = 6). The optimal content of all analytes was achieved under the 70 °C/6-h drying protocol.

### 4.4. Transcriptome Profiling and Co-Expression Network Analysis

Transcriptome analysis commenced with RNA-seq library construction using the Illumina TruSeq Stranded mRNA Library Prep Kit (Illumina, Inc., San Diego, CA, USA), followed by 150 bp paired-end sequencing on the NovaSeq 6000 platform (Illumina, Inc., San Diego, CA, USA).Raw reads underwent rigorous quality control through Trimmomatic v0.39 to eliminate adapter sequences and low-quality bases (Phred score cutoff: 20). High-quality reads were subsequently aligned to an unpublished chromosome-level *A. biserrata* reference genome using HISAT2 v2.2.1, with gene expression quantified as FPKM values via StringTie v2.2.0. Differential expression analysis was performed via DESeq2, applying thresholds of |log2(fold change)| > 1.5 and adjusted *p* < 0.05. Functional enrichment analysis of DEGs was conducted for Gene Ontology (GO) terms and KEGG pathways (clusterProfiler v4.6.2).

Weighted gene co-expression networks were constructed using the WGCNA R package (v1.72) based on FPKM matrices derived from all 33 samples. Following scale-free topology assessment (*R*^2^ > 0.85), a soft-thresholding power of β = 4 was selected to build signed topological overlap matrices. Hierarchical clustering and dynamic tree-cutting identified co-expression modules (minimum size: 200 genes). Module–trait correlations were specifically calculated for Osthole content levels quantified via HPLC (significance threshold: |correlation| > 0.7, *p* < 0.05), alongside tissue types and seasonal timepoints. Hub genes were extracted by intersecting high module membership (MM > 0.9) and gene significance (GS > 0.8) metrics. Osthole-associated modules underwent functional annotation through integrated KEGG pathway and InterPro domain analysis, with putative molecular roles in coumarin biosynthesis assessed via qPCR on selected 30 hub genes.

### 4.5. Statistical Analysis of Metabolite Variation and Sample Classification

Statistical analyses were conducted to compare six metabolite levels across distinct plant parts using independent samples *t*-tests with significance defined at *p* < 0.05, while variation magnitude was quantified via the MAX-MIN Ratio. Principal Component Analysis (PCA) was applied to root samples from different raw material sources to resolve metabolic heterogeneity patterns. Discriminatory metabolites were screened through Coefficient of Variation (CV) assessment, where compounds exhibiting CV > 30% across samples were considered highly variable and thus potentially effective markers for material distinctness.

### 4.6. qRT-Pcr Validation

To validate transcriptome results, 30 key DEGs were selected for quantitative reverse transcription PCR (qRT-PCR). First-strand cDNA was synthesized from 1 μg of total RNA using the PrimeScript™ RT Reagent Kit (Takara, Kusatsu, Japan). Gene-specific primers (Appendix A) were designed with Primer Premier 6.0 and synthesized by Sangon Biotech (Shanghai, China). Reactions were performed on a Bio-Rad CFX96 system with SYBR Green Master Mix (Roche, Basel, Switzerland) under the following conditions: initial denaturation at 95 °C for 5 min, followed by 40 cycles of denaturation at 95 °C for 5 s and annealing/extension at 60 °C for 30 s, with a final extension at 72 °C for 5 min. Melt curve analysis confirmed amplification specificity. Relative expression levels were calculated using the 2^−ΔΔCt^ method with *Rubisco* as the internal control. Statistical significance was assessed via a one-way ANOVA (*p* < 0.05).

## 5. Conclusions

Collectively, this study bridges critical knowledge gaps in *A. biserrata* industrial cultivation through the following validated findings: We reveal an 11-fold enrichment of Osthole in root phloem relative to xylem tissues (*p* < 0.001), suggesting the possible industrial merit of epidermal skin harvesting protocols to maximize bioactive yield. Temporal quantification identified a 340% elevation in Osthole concentration during December versus September harvesting (*p* = 1.8 × 10^−8^), with a further 170% surge following frost events (*p* = 9 × 10^−6^), suggesting mid–late December post-frost as a potentially optimal agricultural window for maximal productivity. Germplasm screening identified elite accession AB-222 exhibiting 230% higher Osthole content versus regional population means, identifying this variant as a promising breeding material for yield stabilization. Weighted gene co-expression network analysis (WGCNA) delineated a gene module strongly correlating with biosynthesis (module-trait r = 0.81, *p* = 6 × 10^−4^) and identified a possible *OMT1* homolog (*AB04G05077*) displaying robust correlation with Osthole accumulation (|R| = 0.88, *p* < 0.001), suggesting its potential utility as a molecular marker for precision breeding.

These integrated discoveries demonstrate a robust, constitutive association between sustained *AbOMT1* gene expression patterns and elevated Osthole content across geographically distinct populations and developmental stages. Importantly, while environmental factors like frost exposure influence biosynthesis, our findings suggest that genetic regulation of *AbOMT1* expression, rather than short-term abiotic triggers alone, is the primary driver defining the observed chemotype persistence in high-Osthole (H) and low-Osthole (L) variants.

Future studies should investigate the role of prolonged field-based frost exposure in modulating genetic pathways controlling *AbOMT1* to account for its ecological significance in natural habitats. Thus, targeted chemotype selection based on stable *AbOMT1* expression markers offers immediate potential for optimizing cultivation programs. Integrating H-chemotype germplasm into breeding strategies will accelerate the development of high-yielding, quality-controlled medicinal material for the reliable production of standardized, Osthole-rich botanical extracts.

## Figures and Tables

**Figure 1 ijms-26-10746-f001:**
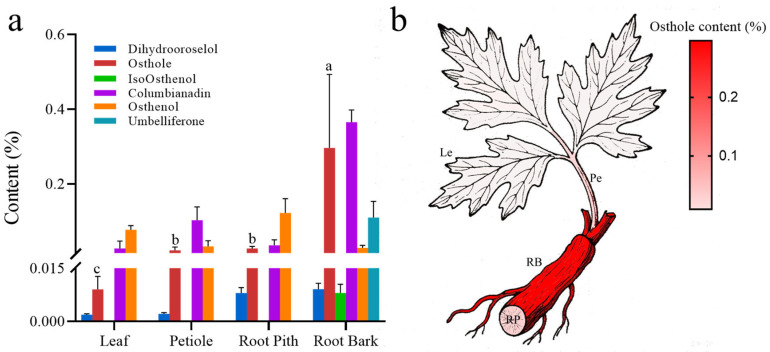
Hierarchical accumulation of Osthole in *A. biserrat* tissues. (**a**) Tissue-specific distribution patterns of six bioactive metabolites in four tissue types (RB: root bark; RP: root pith; Pe: petiole; Le: leaf) from accessions P1, P2, and P3, quantified by UPLC-MS/MS. Data are presented as mean ± SEM. Different letters (a, b, c) above the Osthole bars indicate statistically significant differences between tissue types (*p* < 0.05), with ‘a’ representing the highest content and ‘c’ representing the lowest. (**b**) Spatial concentration gradient of Osthole quantified in four organ compartments, showing mean values from accessions P1, P2, and P3. RB: root bark; RP: root pith; Pe: petiole; Le: leaf.

**Figure 2 ijms-26-10746-f002:**
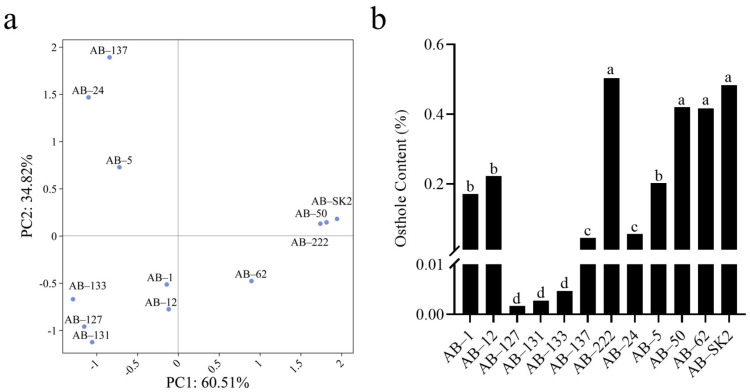
Multivariate analysis of chemotypic variation in *A. biserrata* accessions. (**a**) PCA score plot conducted on six key metabolites across 12 accessions. Axes denote principal components. (**b**) Osthole concentration in the root tissues of different *A. biserrata* accessions. Different letters (a, b, c, d) above bars indicate statistically significant differences between accessions as determined by one-way ANOVA followed by Tukey’s HSD test (*p* < 0.05). Accessions sharing the same letter are not significantly different from each other, with ‘a’ representing the highest Osthole concentration group and ‘d’ representing the lowest.

**Figure 3 ijms-26-10746-f003:**
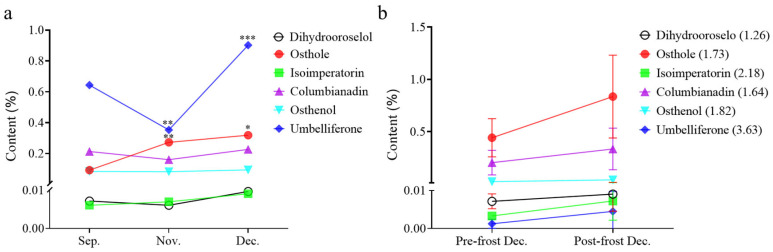
Coumarin metabolite dynamics in *A. biserrata* root. (**a**) Temporal variations in six coumarin metabolites across developmental stages (September, November, and December) in accessions AB-182. Asterisks indicate significant differences compared to the previous sampling time point (* *p* < 0.05, ** *p* < 0.01, *** *p* < 0.001; one-way ANOVA followed by Tukey’s post-hoc test). (**b**) Changes in six coumarin metabolite levels across nine *A. biserrata* accessions sampled (AB-3, AB-5, AB-25, AB-55, AB-60, AB-67, AB-69, AB-70, AB-92) within 24 h before and after the frost event in December. Values in parentheses indicate fold-change in metabolite concentration post-frost relative to pre-frost levels (post-frost/pre-frost ratio). Data are presented as mean ± SEM.

**Figure 4 ijms-26-10746-f004:**
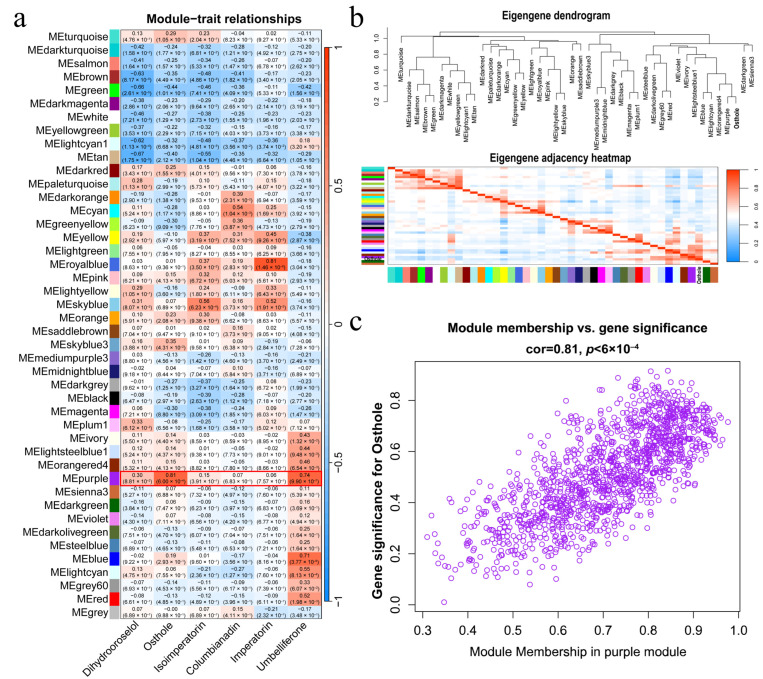
Network analysis of coumarin biosynthesis modules. (**a**) Heatmap of correlation coefficients between key coumarin metabolites and co-expression modules. Pairwise Pearson *r* values with adjusted *p*-values (Benjamini–Hochberg) are numerically overlaid. The purple module (highlighted with a red box) shows the highest correlation with Osthole. (**b**) Bivariate clustering analysis showing module associations with Osthole accumulation. Top: hierarchical clustering dendrogram of modules based on Osthole correlation profiles; bottom: module–trait relationship heatmap. The color scale from red (positive correlation) to blue (negative correlation) indicates the strength and direction of correlations between modules and Osthole content. (**c**) Scatter plot validating hub genes in the purple module. Module membership (MM, *x*-axis) vs. gene significance for Osthole (GS, *y*-axis).

**Figure 5 ijms-26-10746-f005:**
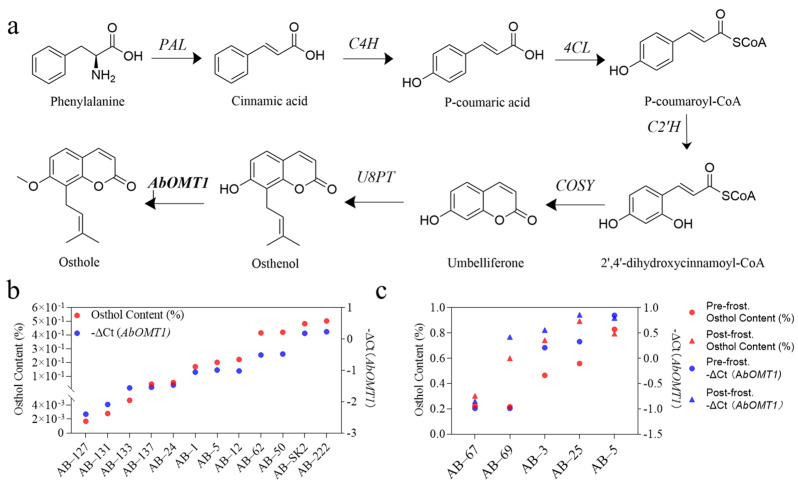
Functional validation of *AbOMT1* as a key determinant of Osthole biosynthesis. (**a**) Proposed biosynthetic pathway from phenylalanine to Osthole. (**b**) Correlation between Osthole content (%) and *AbOMT1* expression levels (measured as −ΔCt) across 12 diverse *A. biserrata* accessions. (**c**) Relationship between Osthole accumulation (%) and *AbOMT1* transcript abundance (−ΔCt) in five accessions before and after frost exposure.

## Data Availability

The raw RNA-seq data generated in this study have been deposited in the NCBI under accession number PRJNA1312167.

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
