# Peer review of "Spatiotemporal Distribution Patterns of Osthole and Expression Correlation of the MOT1 Homologue in Cultivated Angelica biserrata"

_ijms, 2025, doi:10.3390/ijms262110746_

Round 1

Reviewer 1 Report

Comments and Suggestions for Authors

The work can be interested in that that admissions the amount of kumarin in different tissues  and species. this can be valuable for further breeding.  transcriptomic analysis and further speculations should be not overinterpreted and treated only as predictions and speculations. For further evaluation of the paper many additional data are required which authors does not show. Additional experiments can be required.

Fig 1a. - Instead Present tissues and amounts of metabolites. 1b - add datatable.

These accessions consistently demonstrated the highest osthol accumulation. - show data on this, not CV.

Figure 3a. Only tree time points ? explain why you did not measure in earlier time. Add more timepoints. 

Figure 3b. Give additionally relative values. 24 h pre- and post… - here add more timepoints 2,5,10 days after frost. 

Table 5S -> Raw counts, + full description of all samples. 

gene  AbOMT1  191 

(AB04G05077), the homologue of OMT1 (METABOLIC O-METHYLTRANSFERASE 1),  192 

emerged as the top-ranked hub gene, demonstrating a robust correlation with Osthole  193 

content traits (

  • Show the expression values and metabolite values to see the correlation.

KEGG enrichment analyses revealed .. - use other modules as negative controls, present results. 

Validation via qRT-PCR using root samples representing diferent tissues,  - show this data

Fig 5a Biosynthetic pathway from phenylalanine to Osthole.  - it is not your results, remove, or explain how it relates to you results. 

Everywhere on figs: add accession numbers, genes, tissues, timepoints etc. 

Comments on the Quality of English Language

The English must be checked by professional editing service or by the Professional. 

Author Response

The work can be interested in that that admissions the amount of kumarin in different tissues  and species. this can be valuable for further breeding.  transcriptomic analysis and further speculations should be not overinterpreted and treated only as predictions and speculations. For further evaluation of the paper many additional data are required which authors does not show. Additional experiments can be required.

Comments 1: Fig 1a. - Instead Present tissues and amounts of metabolites. 1b - add datatable.

Response 1: Thank you for pointing this out. We have revised Figure 1a: a bar chart showing the contents of different metabolites in different tissues of the three accessions has been created as the new Figure 1a, which can also serve as a data reference for Figure 1b. The original data of the former Figure 1a is provided in Table S2.

Comments 2: These accessions consistently demonstrated the highest osthol accumulation. - show data on this, not CV.

Response 2: Thank you for pointing this out. We have redrawn Figure 2b: the osthole contents of different accessions are presented as a bar chart in the new Figure 2b. The CV (Coefficient of Variation) chart from the original Figure 2b has been moved to the supplementary materials as Supplementary Figure S1. We conducted the CV analysis to illustrate that osthole can serve as a key indicator for germplasm screening.

Comments 3: Figure 3a. Only tree time points ? explain why you did not measure in earlier time. Add more timepoints.

Response 3: Thank you for pointing this out. The three sampling points (September, November, December) were strategically selected to capture key developmental stages in A. biserrata's biennial growth cycle. September represents the early dormancy phase when vegetative growth ceases and resources begin redirecting to the rhizome. November captures mid-dormancy when secondary metabolite accumulation typically intensifies in perennial medicinal plants. December sampling coincided with deep dormancy following frost exposure (now LINE 142-149).

Earlier time points (e.g., summer months) were not included because A. biserrata primarily allocates resources to aerial growth during that period, with minimal secondary metabolite production in roots - a pattern well-documented in Apiaceae perennials. Our preliminary screenings of summer samples showed negligible coumarin levels that fell below reliable quantification thresholds for several compounds. Additionally, traditional harvesting for medicinal purposes occurs exclusively during dormancy months, making these time points most relevant for pharmacological applications (now LINE 142-149).

In future studies, we plan to implement a more comprehensive temporal analysis spanning the complete biennial cycle, including early developmental stages.

Comments 4: Figure 3b. Give additionally relative values. 24 h pre- and post… - here add more timepoints 2,5,10 days after frost.

Response 4: Thank you for pointing this out. The 24-hour pre- and post-frost comparison was initially opportunistic rather than planned - an unexpected early frost event presented a rare research opportunity to capture immediate metabolic responses. When we observed the dramatic short-term increase in Osthole concentration, we collected these samples to document this previously unreported phenomenon (now LINE 154-160).

While we fully agree that extended time-course sampling (2, 5, 10 days post-frost) would provide valuable insights into the sustainability of this response, logistical constraints prevented immediate follow-up sampling. Specifically, persistent sub-freezing temperatures following the initial frost event made field access and sample preservation challenging without introducing experimental artifacts (now LINE 154-160).

Comments 5: Table 5S -> Raw counts, + full description of all samples.

Response 5: Thank you for pointing this out. Table S5 has been revised: the original FPKM values have been replaced with Count. Detailed information for each sample has been listed in Tables S1, S3, and S4, so it is no longer included in Table S5.

Comments 6: gene AbOMT1 191(AB04G05077), the homologue of OMT1 (METABOLIC O-METHYLTRANSFERASE 1), 192emerged as the top-ranked hub gene, demonstrating a robust correlation with Osthole 193 content traits (

Show the expression values and metabolite values to see the correlation.

KEGG enrichment analyses revealed .. - use other modules as negative controls, present results.

Validation via qRT-PCR using root samples representing diferent tissues, - show this data

Response 6: Thank you for pointing this out. The metabolite content data of AbOMT1 and osthole have been listed in the now Table S8, and also presented in the revised Figure 5b-c as well as now Figure S4b.

The GO and KEGG enrichment results of genes in the gray module (which show extremely low correlation with osthole) have been added to now Figure S3c-d as controls (now LINE 223-227).

The data for validating transcriptomic results via qRT-PCR have been included in now Table S7, and also displayed in now Figure S4a.

Comments 7: Fig 5a Biosynthetic pathway from phenylalanine to Osthole. - it is not your results, remove, or explain how it relates to you results.

Response 7: Thank you for pointing this out. Figure 5a is a diagram of the osthole biosynthetic pathway. It is primarily intended to correspond to the WGCNA results stated in the section: "Key co-expressed genes within the module included: prenyltransferases (AB10G00842, AB06G01999, AB07G04187, AB11G02263); COUMARIN SYNTHASE (COSY) regulating umbelliferone biosynthesis (AB01G01375, AB06G01114); and core phenylpropanoid biosynthetic genes (PAL, C4H, 4CL, C2H). All these genes are key genes in the pathway of osthole biosynthesis from phenylalanine (Figure 5a)." These genes are located in the purple module and all play important roles in the osthole biosynthetic pathway. Meanwhile, Figure 5a also shows the position of the AbOMT1 gene in the osthole biosynthetic pathway (now LINE 210-215).

Comments 8: Everywhere on figs: add accession numbers, genes, tissues, timepoints etc.

Response 8: Thank you for pointing this out. The figures and their figure legends have been revised, with the corresponding information added.

Comments 9: The English could be improved to more clearly express the research. Figures and tables must be improved.

Response 9: Thank you for pointing this out. We have thoroughly revised the manuscript to improve the English expression throughout the text, enhancing clarity and precision. Additionally, all figures and tables have been refined with improved resolution, clearer labeling, and more informative captions. We believe these revisions have significantly enhanced the quality and accessibility of our work.

Reviewer 2 Report

Comments and Suggestions for Authors

I express my deepest admiration to the authors for their superbly planned and flawlessly executed work. This paper contributes new data to the study of the metabolism and accumulation of pharmacologically valuable coumarins and provides a foundation for future research in the field of medicinal plant physiology. I read the manuscript with great pleasure; there were a few difficult points to understand, but the methods section and the accompanying supporting materials answered all my questions. I found only a few typos and minor comments that I would recommend correction.

Line 18 has typo “osthoe”

In the introduction, at the first mention of the full name of the species Angelica biserrata, the authors (R.H.Shan & Yuan) C.Q.Yuan & R.H.Shan should be added (Angelica biserrata (R.H.Shan & Yuan) based on - https://wfoplantlist.org/taxon/wfo-0000536120-2025-06 )

Move the conclusions from the last paragraph of the introduction to the results or conclusion section, and rewrite this paragraph as the purpose and objectives of the paper. It feels a bit odd to read the results in the introduction.

Section 4.3 describes gradient elution. For example, “The mobile phase consisted of a gradient elution of 0.1 % aqueous acetic acid (A) and acetonitrile (B). Binary gradient with A and B was installed starting with 0% B. The following gradient profile was used: 0-10 min linear gradient from 0% to 15 % B; 10-25 min isocratic on 15 % B; 25-40 min linear gradient from 15% to 50 % B; 40-55 min isocratic on 50 % B; 55-65 min linear gradient from 50% to 60 % B; 65-80 min isocratic on 60 % B; 80-100 min linear gradient from 60% to 100 % B.”

Line 367 “Qrt-Pcr Validation” change to “qRT-PCR Validation”

Line 251 “Abomt” change to “AbOMT”

Lines 233 “Apiaceae” write in italics

Best of luck in your research and look forward to reading future publications,

Reviewer

Author Response

I express my deepest admiration to the authors for their superbly planned and flawlessly executed work. This paper contributes new data to the study of the metabolism and accumulation of pharmacologically valuable coumarins and provides a foundation for future research in the field of medicinal plant physiology. I read the manuscript with great pleasure; there were a few difficult points to understand, but the methods section and the accompanying supporting materials answered all my questions. I found only a few typos and minor comments that I would recommend correction.

Comments 1: Line 18 has typo “osthoe”

Response 1: Thank you for pointing this out. Revisions have been made in the manuscript (now LINE 19).

Comments 2: In the introduction, at the first mention of the full name of the species Angelica biserrata, the authors (R.H.Shan & Yuan) C.Q.Yuan & R.H.Shan should be added (Angelica biserrata (R.H.Shan & Yuan) based on - https://wfoplantlist.org/taxon/wfo-0000536120-2025-06 )

Response 2: Thank you for pointing this out. Revisions have been made in the manuscript (now LINE 32).

Comments 3: Move the conclusions from the last paragraph of the introduction to the results or conclusion section, and rewrite this paragraph as the purpose and objectives of the paper. It feels a bit odd to read the results in the introduction.

Response 3: Thank you for pointing this out. The last paragraph of the Introduction has been rewritten (now LINE 87-97). Additionally, this rewritten last paragraph of the Introduction has been placed at the beginning of the Conclusion section, and the entire Conclusion section (including this integrated paragraph) has been revised to enhance its readability and fluency (now LINE 402-415).

Comments 4: Section 4.3 describes gradient elution. For example, “The mobile phase consisted of a gradient elution of 0.1 % aqueous acetic acid (A) and acetonitrile (B). Binary gradient with A and B was installed starting with 0% B. The following gradient profile was used: 0-10 min linear gradient from 0% to 15 % B; 10-25 min isocratic on 15 % B; 25-40 min linear gradient from 15% to 50 % B; 40-55 min isocratic on 50 % B; 55-65 min linear gradient from 50% to 60 % B; 65-80 min isocratic on 60 % B; 80-100 min linear gradient from 60% to 100 % B.”

Response 4: Thank you for pointing this out. We have rewritten Section 4.3, adding specific details and procedures for operations and elution (now LINE 344-358).

Comments 5: Line 367 “Qrt-Pcr Validation” change to “qRT-PCR Validation”

Response 5: Thank you for pointing this out. Revisions have been made in the manuscript (now LINE 390).

Comments 6: Line 251 “Abomt” change to “AbOMT”

Response 6: Thank you for pointing this out. Revisions have been made in the manuscript (now LINE 271).

Comments 7: Lines 233 “Apiaceae” write in italics

Response 7: Thank you for pointing this out. Revisions have been made in the manuscript (now LINE 254).

Round 2

Reviewer 1 Report

Comments and Suggestions for Authors

Generally, all points are constructively responded, I am still not very happy with overinterpretation of expression networks results, but it is acceptable now, only add “possible” etc. 

Purple module - color is not a biological result, write  - identified a module (colored purple fig … ) , … 

See the suggested/corrected Abstract.

I suggest minor revision, so Authors can add the above changes. 

Abstract

Sustainable cultivation of Angelica biserrata, a medicinal specie with a bioactive coumarin 

Osthole, is hindered by an inconsistent metabolite accumulation. To address this limitation, we integrated spatiotemporal metabolomics and transcriptomic analyses. Tissue-specific measurements revealed that root bark accumulates Osthole at 0.30 ± 0.15%, a concentration 11-fold higher compared to root pith 

and 15–30-fold higher compared to aerial organs. In time, the Osthole content increased by 195% from 

September to December, with a frost exposure further increasing the accumulation by additionally 170%. Germplasm 

screening identiied an elite accession, AB-222, exhibiting a 230% higher Osthole content compared 

to regional averages. Weighted gene co-expression network analysis identified a gene module, 

strongly correlating with Osthole accumulation. Within this module, AbOMT1 (AB04G05077), an O-

METHYLTRANSFERASE 1 (OMT1) homolog encoding an S-adenosyl methionine-dependent O-me-

thyltransferase, was the top hub gene. AbOMT1 expression reflected Osthole dynamics both spa-

tially (three-fold higher in root bark vs. root pith) and temporally. Module functional analysis re-

vealed significant enrichment in phenylpropanoid and monoterpenoid biosynthesis pathways. 

Our results suggest AbOMT1 as a possible key molecular marker for Osthole accumulation, establish frost induction as a strong yield regulator, and suggest AB-222 as an elite germplasm resource. 

Author Response

Comments 1: Generally, all points are constructively responded, I am still not very happy with overinterpretation of expression networks results, but it is acceptable now, only add “possible” etc. 

Purple module - color is not a biological result, write  - identified a module (colored purple fig … ) , … 

See the suggested/corrected Abstract.

I suggest minor revision, so Authors can add the above changes. 

Response 1: We sincerely appreciate the thoughtful and constructive feedback provided on our manuscript. We have carefully addressed all concerns, particularly regarding the interpretation of expression network results and the presentation of module colors as biological outcomes. Below, we outline the specific changes made to improve the scientific accuracy and cautious interpretation of our findings:
1. Network Analysis Presentation: We have revised our descriptions of co-expression modules throughout the manuscript (lines 181-183, 202-204, 288-289, 413-414) to avoid presenting color designations as biological results. For example, we now describe "a gene module strongly correlating with Osthole accumulation" rather than referring to the "purple module" as a biological entity.
2. Measured Language: We have incorporated more cautious terminology by adding qualifiers like "possible," "potential," and "suggests" in key sections discussing experimental findings (lines 78, 81, 104-105, 154, 182, 206, 273, 308, 409, 419-420). This addresses the concern about overinterpretation of expression network results.
3. Moderated Claims: We have revised statements that might have appeared overly conclusive (lines 411-412, 415-416, 420) by replacing definitive terms like "establishing" with more measured language such as "suggesting the possible" and "identified a possible" instead of "uncovered."
4. Scientific Precision: We have improved scientific precision throughout the manuscript (lines 239, 405, 414) by focusing on functional descriptions rather than visual representations in our data analysis.
5. Abstract Revision: We have completely replaced our original abstract with the reviewer-suggested version. We appreciate this valuable contribution as it presents our findings in a more measured and scientifically accurate manner, properly contextualizing the significance of our work while avoiding overstatements.
Thank you for your valuable feedback that has helped improve the quality of our manuscript. We hope these revisions adequately address your concerns.